# AirSign: Smartphone Authentication by Signing in the Air

**DOI:** 10.3390/s21010104

**Published:** 2020-12-26

**Authors:** Yubo Shao, Tinghan Yang, He Wang, Jianzhu Ma

**Affiliations:** Department of Computer Science, Purdue University, West Lafayette, IN 47907, USA; yang1683@purdue.edu (T.Y.); hw@purdue.edu (H.W.); majianzhu@purdue.edu (J.M.)

**Keywords:** security and privacy protection, biometrics, acoustic sensing, online signature authentication, machine learning

## Abstract

In this paper, we propose AirSign, a novel user authentication technology to provide users with more convenient, intuitive, and secure ways of interacting with smartphones in daily settings. AirSign leverages both acoustic and motion sensors for user authentication by signing signatures in the air through smartphones without requiring any special hardware. This technology actively transmits inaudible acoustic signals from the earpiece speaker, receives echoes back through both built-in microphones to “illuminate” signature and hand geometry, and authenticates users according to the unique features extracted from echoes and motion sensors. To evaluate our system, we collected registered, genuine, and forged signatures from 30 participants, and by applying AirSign on the above dataset, we were able to successfully distinguish between genuine and forged signatures with a 97.1% F-score while requesting only seven signatures during the registration phase.

## 1. Introduction

With the rapid development of wireless networking and mobile computing technology, reliable user authentication technologies are urgently needed for smartphone users to protect their valuable information. People tend to spend more time and preserve their private and sensitive information, such as social networks [1,2] and online banking accounts, on their smartphones. Currently, many user authentication technologies have been used in today’s smartphones. However, each of the existing technologies has its limitations.

Password (personal identification number (PIN)) authentication [3] is the most basic and traditional of these technologies. In reality, people are more likely to use shorter passwords, which are easier for a shoulder surfer to acquire, and therefore make this method less safe [4]. Fingerprint authentication [5,6] is another method that has been applied to most smartphones nowadays. Nevertheless, its accuracy is severely affected by the state of one’s fingers (e.g., it can make mistakes with the dryness or the cleanness of the fingers’ skin [7]). Additionally, other people could put a user’s finger on his/her smartphone screen while he/she is unconscious, which would make this authentication method easy to attack. More advanced fingerprint authentication uses ultrasonic signals [6] to capture the unique 3D characteristics of the user’s fingerprint, but it requires significant hardware changes. Apple’s FaceID [8] is a state-of-the-art way to securely authenticate users. However, FaceID requires special sensors, such as a dot projector, a flood illuminator, and an infrared depth sensor, which require extra screen space and hardware cost. Therefore, considering the limitations of the above existing authentication methods, we are interested in asking: Is it possible to develop a user authentication technology (1) with higher authentication accuracy; (2) with no extra hardware; (3) that could not easily be cracked by shoulder surfers; (4) that is easier and more flexible to use?

In this paper, we aim at leveraging the existing sensors on smartphones to develop a novel user authentication technology, AirSign, which allows users to accomplish the authentication process by signing their signatures in the air. As Figure 1 shows, smartphone users can use one of their hands to create signatures in the air without touching the screen while the other hand is holding the smartphone. The main idea behind our approach is to exploit the potential of both acoustic and motion sensors to extract unique features of different users and distinguish them according to these features. In this work, we make a basic assumption that the human body will move as an entire entity when signing in the air in front of their smartphone. Our assumption originates from the inner workings of the human brain. Corballis et al. [9] shows that the left hemisphere controls the right half of the body, and the right hemisphere controls the left half of the body. Bundy et al. [10] shows that both sides of the brain are active during one-sided arm movement. They also show that the same side of the brain mirrors the neural activity of the opposite hemisphere in the context of reaching movements. Thus, when the user is signing their signatures in the air using his/her right hand, the left hemisphere controls this movement, but the right hemisphere mirrors the neural activity of the left hemisphere. If so, the left hand holding the smartphone will receive the same neural signals as the right hand moves and triggers mild movement. Hence, the motion captured by the left hand could be considered as the behavioral characteristics of our signature data. Specifically, when a user is signing his/her signature in the air, acoustic signals are transmitted by the earpiece speaker of the smartphone; they travel through the air, reach the user’s hand, and reflect back to both the top and bottom microphones.

Since the movement features of different people’s signing processes are different, it is possible to distinguish them according to their reflected sound signals. In addition, motion sensors, including the accelerometer and gyroscope sensors, can continuously capture the features of acceleration and rotation of the smartphone during the signing process, since one of the user’s hands that holds the smartphone cannot be perfectly motionless. Next, using all these extracted features, the Multidimensional Dynamic Time Warping (MD-DTW) method [11] is adopted to calculate the similarity between the registered signatures and the new signature. Users may unlock their devices if the similarity is less than a specific threshold. Additionally, to improve the system performance, we add a hand geometry authentication module, which could recognize different people by the features of their hands and fingers, including hand size, finger length, and distance from the screen. In this step, before a user starts signing his/her signature, he/she is required to hold the hand in front of the smartphone screen for 0.5 s, during which time the acoustic sensors emit specially designed sound waves and collect and extract features of the reflected signals. The user could sign his/her signature over the smartphone screen in a three-dimensional space of 25 cm × 25 cm and 5–15 cm hovering over the earpiece speaker. The details of our system will be discussed in Section 3.

People may ask why they should sign their signatures in the air instead of on the smartphone screen. There are several reasons. First of all, people’s fingers are not limited in the space of a small screen when they are signing their signatures in the air, which will make users feel more comfortable. Second, when people are signing their signatures on the screen, it is hard for the authentication system to leverage the acoustic sensors because only a tiny part of the sound signals transmitted by the earpiece speaker can be reflected back to the built-in microphones. Without the help of acoustic sensors, the usability of our system will decrease. Last but not least, it is relatively easy to imitate a user’s signature on the screen by watching how he/she signs because the screen is only a two-dimensional space. However, it is hard to imitate a three-dimensional signature in the air. From this perspective, signing signatures in the air is more secure.

In order to evaluate our system, we applied it on registered, genuine, and forged signatures collected from 30 participants. During the enrollment phase when users were registering their signatures on smartphones, we assumed that attackers could not have physical access to these devices. After the enrollment phase, we assumed that attackers could have physical access to people’s smartphones in the following ways: stealing devices or picking up lost devices. In addition, we assumed that the attackers could launch shoulder surfing attacks when users were signing their signatures on smartphones. The details of how the above signature data were collected for our system will be discussed in Section 4.

The contributions of our paper can be summarized as follows:We designed a smartphone signature authentication system that allows users to sign their signatures in the air, which improves the convenience and flexibility of the signing authentication process.We, for the first time, leveraged both acoustic sensors and motion sensors on a smartphone to detect different users’ hand geometries, trace their signing processes in the air, and extract essential features to verify their identities.We implemented a smartphone prototype application and collected user surveys to evaluate our AirSign system. The evaluation demonstrated that our system can authenticate users with an F-score of 97.1%.

The rest of this paper is organized as follows. In Section 2, we give a brief introduction to the system architecture. In Section 3, we discuss the detailed design of the proposed AirSign system. In Section 4, we describe the experiments that we conducted to collect registered, genuine, and forged signatures. In Section 5, we evaluate the proposed system from different perspectives. Section 6 provides the relevant background and related work. Section 7 discusses the limitations and future work for our system. Section 8 summarizes this paper.

## 2. Overview

The AirSign system leverages two types of sensing modules to implement the authentication process of signing signatures in the air. Acoustic sensing is one type that uses the built-in speaker and microphones. Another type is motion sensing, which uses the accelerometer and gyroscope sensors. Figure 2 demonstrates an overview of the system architecture, which contains two phases, i.e., a user enrollment phase and a user authentication phase.

### 2.1. User Enrollment Phase

In the enrollment step, each user is required to register his/her signature for *N* times. Now, we are going to discuss how acoustic and motion sensors work together to collect registration data and extract each user’s unique features in the following description.

#### 2.1.1. Acoustic Sensing

The acoustic sensing includes two modules—hand geometry sensing and signature sensing. At the beginning of registration, each user is required to hold the smartphone with one hand and keep the other hand in front of the screen for around 2 s. At the same time, the earpiece speaker emits pre-designed chirp sound signals, which will reach the user’s hand and reflect back to the top and bottom microphones. Then, our AirSign system will extract the user’s hand geometry features by calculating the spectrogram of the reflected sound waves. When a user starts signing his/her signature, the earpiece speaker begins emitting sound signals with different fixed frequencies. In a way similar to [12], we can use the echoes to acquire in-air signature features, including displacement, velocity, and acceleration, during the signing process.

#### 2.1.2. Motion Sensing

The motion sensing happens when the user starts signing his/her signature. Since the hand that holds the phone will inevitably move slightly, and the same person tends to move the “holding hand” with a similar pattern, the motion sensors, including the accelerometer and gyroscope sensors, can capture the specific features of this movement, such as acceleration and angular velocity.

The combination of those features extracted in each registration, including features of hand geometry, signature, and motion, will be regarded as one “Air Signature”. After collecting the user’s registered data, we use them to train three types of classifiers, i.e., a hand geometry classifier, signature classifier, and motion classifier, which will be used for the future decision (genuine or forged) for signature authentication.

### 2.2. User Authentication Phase

In the authentication step, a specific user needs to follow a similar procedure to that of the enrollment step: putting one hand in front of the smartphone for around 0.5 s and creating his/her signature. Then, the features extracted by acoustic and motion sensors will be created as a new “Air Signature”. Next, our system will calculate the dissimilarity between the new “Air Signature” and the registered ones using the three trained classifiers, and decide whether it is genuine or forged data.

## 3. System Design

In this section, we are going to discuss the detailed design of the AirSign system, which is divided into three different parts: sound signal design, feature extraction, and decision model.

### 3.1. Sound Signal Design

#### 3.1.1. Selection of Acoustic Sensors

There are typically two speakers and two microphones on a smartphone. In terms of speakers, the top one is mainly used for phone calls, while the bottom one can provide a better sound effect. With regard to microphones, the main one is usually at the bottom for phone calls, and the other one is at the top for noise cancellation. See Figure 1 for more details.

In the proposed AirSign system, because of the location of the bottom speaker, the energy of the emitted sound signal will focus on the downward direction. In other words, the amount of energy that reaches users’ fingers will be small, and the reflected signal may be influenced by the directed signal, causing inaccurate acoustic data. Therefore, we only use the earpiece speaker in our system. During the signing process, since we need to capture the information about how users’ fingers move in the air, both the top and bottom built-in microphones should be used to acquire the movement data.

#### 3.1.2. Overview of Sound Signal Design

We discuss whether the features extracted by acoustic sensors are accurate enough to represent the user’s hand geometry and the signature has a huge influence on the final performance of AirSign system. Therefore, there are several factors that we need to consider in the design of sound signals:To avoid disturbances to other people, the acoustic signals should be inaudible. According to [13], the sound above 17 kHz is difficult to hear. On the other hand, the highest sampling rate on a typical smartphone is 48 kHz, which means that the highest frequency of the sound wave should not be greater than 48 kHz/2 = 24 kHz to avoid aliasing. After extensive experiments on analyzing different frequency intervals, we choose 20–23 kHz as the frequency interval for both the hand geometry phase and the signature phase.To ensure that echoes from the hand geometry section do not overlap with the transmitted signal from the in-air signature section, we need to provide a gap between two sections. After testing different distance ranges for the above signal, we find that the echoes reflected by objects are very weak if the distance between objects and reflectors is over 3 m. Thus the maximum delay can be calculated as (2 × 3 m) /(343 m/s) = 17.5 ms. In AirSign, we add a little buffer and set the time interval between the hand geometry section and the in-air signature section to be 20 ms. An illustrative example of the acoustic signals is shown below in Figure 3.To increase the signal-to-noise ratio (SNR) and prevent frequency echo leaks in both hand geometry and in-air signature sections, a Hanning window [14] is applied to reshape all emitted sound waves.

#### 3.1.3. Sound Signal for Hand Geometry

As introduced in the above subsection, the hand geometry section is transmitted before the in-air signature section. During the hand geometry section, a user holds his/her hand or finger in front of the smartphone screen for a short period. The speaker emits 20–23 kHz chirp waves in which the frequency increases linearly with time. As shown in Figure 4, a segment of the chirp signal is emitted every 20 ms, while each one only lasts for 1 ms. This design is to make sure that the emitted signals will not overlap with the echoes from the previous chirp wave. Since the features of hand geometry, including hand size, finger length, distance from the screen, etc., are quite different among users, we expect that these features are encoded in the reflected signals and can be extracted to distinguish different users.

#### 3.1.4. Sound Signal for Signature

During the signing process, to sense the users’ signing trace using the Doppler effect, we transmit a continuous cosine sound wave, Acos2πft, in which *A* denotes the amplitude and *f* denotes the frequency of the sound signal. However, due to the multi-path effect of the sound signal that is reflected by the user’s finger, hand, and the environment, obtaining an accurate phase length change becomes difficult when the user is signing his/her signature in the air. We adopt frequency diversity to solve this problem [12]. More specifically, to measure the accurate path length change, a continuous sound wave that contains *n* different frequencies ranging from 20 kHz to 23 kHz with a step of 200 Hz is transmitted. The designed wave can be denoted as $\frac{1}{n}\sum_{i=1}^{n}cos2\pi f_{i}t$.

### 3.2. Feature Extraction

In this subsection, we will discuss how AirSign extracts features of the hand geometry, the signature, and the motion.

#### 3.2.1. Hand Geometry Feature Extraction

The detailed design of the receiver is shown in Figure 5. After both the top and bottom built-in microphones receive the echoes, a Butterworth bandpass filter [15] is adopted to remove the background noise. The direct signal that which travels from the earpiece speaker to the microphones is also needed to be removed. The removal of the direct signal is implemented by calculating the difference between the echoes with and without the signing hand in front of the screen.

The illustration of the relationship between time and frequency is shown in Figure 6. The solid blue line represents the transmitted sound signal, while different dashed red lines represent the echoes of the direct path and multi-paths. In addition, we use different shades to represent the power of echoes. After all of these processing steps, Figure 7 is obtained, in which the echoes coming from the reflection of both the users’ hands and the environment are separated clearly.

For each hand geometry section for the registered user, we first calculate the spectrograms of echoes using short-window Fast Fourier Transform (FFT). Illustrative figures of the spectrograms are shown in Figure 8, while two subfigures of each row represent a specific user. From Figure 8, it can be observed that the differences between the patterns of users are obvious enough to be noticed by our eyes. To better distinguish different users by their spectrograms, a K nearest neighbors (KNN)-based classifier is trained considering the size of training samples and by providing higher recall. Specifically, we regard the spectrograms obtained from the registered user’s hand geometry as positive data of the training set, while randomly selecting the same number of samples from spectrograms obtained from other users’ hand geometry as negative data of the training set. When new hand geometry data comes, the KNN-based model will classify the new data into the category of genuine or forged for a certain user.

#### 3.2.2. Signature Feature Extraction

As mentioned above, we use the top and bottom built-in microphones on the same device to receive the reflected sound signal for the signature in our AirSign system. Similarly to the hand geometry section, the echoes should pass through a Butterworth bandpass filter to remove the background noise. This step is essential, especially for collecting signature data in some noisy environments.

After the removal of background noise, the signal is then split into two identical copies and multiplied by cos2πft and −sin2πft. We then remove high-frequency components and downsample the signal to get in-phase and quadrature components using a cascaded integrator comb (CIC) filter. Once we obtain the in-phase and quadrature signals, the path length change will be calculated through the local extreme value detection (LEVD) algorithm [12]. Next, we combine all the results of the path length change for each frequency and apply linear regression to achieve more accurate path length changes for both top and bottom built-in microphones, denoted as $d_{top}\left(t\right)$ and $d_{bot}\left(t\right)$. Then, a local feature-based approach where the features are derived from each point along the online trace of a signature is adopted. In addition to $d_{top}\left(t\right)$ and $d_{bot}\left(t\right)$, we also consider the following features, such as:First-order differences:
$$\begin{array}{c} \left\{\begin{array}{c} \Delta d_{top}\left(t\right)=d_{top}\left(t+1\right)-d_{top}\left(t\right)\\ \Delta d_{bot}\left(t\right)=d_{bot}\left(t+1\right)-d_{bot}\left(t\right) \end{array}\right. \end{array}$$Second-order differences:
$$\begin{array}{c} \left\{\begin{array}{c} \Delta^{2}d_{top}\left(t\right)=\Delta d_{top}\left(t+1\right)-\Delta d_{top}\left(t\right)\\ \Delta^{2}d_{bot}\left(t\right)=\Delta d_{bot}\left(t+1\right)-\Delta d_{bot}\left(t\right) \end{array}\right. \end{array}$$

In addition, we consider the combination features obtained by both the top and bottom built-in microphones, such as:Sine and cosine features:
$$\begin{array}{c} \left\{\begin{array}{c} \sin\left(t\right)=\frac{\Delta d_{bot}\left(t\right)}{\sqrt{{\left(\Delta d_{top}\left(t\right)\right)}^{2}+{\left(\Delta d_{bot}\left(t\right)\right)}^{2}}}\\ \cos\left(t\right)=\frac{\Delta d_{top}\left(t\right)}{\sqrt{{\left(\Delta d_{top}\left(t\right)\right)}^{2}+{\left(\Delta d_{bot}\left(t\right)\right)}^{2}}} \end{array}\right. \end{array}$$Length-based features:
$$\begin{array}{c} \left\{\begin{array}{c} l\left(t\right)=\sqrt{{\left(\Delta d_{top}\left(t\right)\right)}^{2}+{\left(\Delta d_{bot}\left(t\right)\right)}^{2}}\\ \Delta l\left(t\right)=\sqrt{{\left(\Delta^{2}d_{top}\left(t\right)\right)}^{2}+{\left(\Delta^{2}d_{bot}\left(t\right)\right)}^{2}} \end{array}\right. \end{array}$$
where t=1,2,⋯,n−1 and *n* denote a discrete time point. Finally, the above 10 features are selected, i.e., {$d_{top}\left(t\right)$, $d_{bot}\left(t\right)$, $\Delta d_{top}\left(t\right)$, $\Delta d_{bot}\left(t\right)$, $\Delta^{2}d_{top}\left(t\right)$, $\Delta^{2}d_{bot}\left(t\right)$, sin(t), cos(t), l(t), and Δl(t)}, as extracted features in the signature data. Therefore, a signature classifier that is based on the signature features can be trained.

We compare the displacement features of the registered, the genuine, and the forged signature data obtained by the bottom microphone from one certain user in Figure 9. It can be observed that the genuine data are very similar to the registered data, while the forged data provided by a skilled attacker show much difference.

#### 3.2.3. Motion Feature Extraction

As introduced in the system overview, when a user is signing his/her signature in the air using one hand, the other hand that holds the smartphone could be observed moving slightly. Our in-air signing process includes two components: the signing action of the right hand (assuming the user signs with the right hand), and the movement of the left hand, which holds the smartphone. One important insight of our work is that we consider the human body as an entire entity. That is, the movement of the right (signing) hand will drive the movement of the left (holding) hand through the body. Another important insight is the coupling of the acoustic features with the motion features of the users. It is relatively easy for someone to imitate the motion of signing, but it is typically hard to imitate the strength and speed of the signing, which are greatly reflected by the acoustic features. Therefore, motion sensors, such as the accelerometer and gyroscope sensors on smartphones, can be applied to continuously capture the displacement and rotation of the device for the holding hand. We use $\left\{Acc_{x}\left(t\right),Acc_{y}\left(t\right),Acc_{z}\left(t\right)\right\}$ to denote the three-dimensional linear acceleration from the accelerometer, and use $\{Gyro_{x}\left(t\right)$, $Gyro_{y}\left(t\right),$ and $Gyro_{z}\left(t\right)\}$ to denote the angular velocity from the gyroscope. We also consider the above angular velocity’s first-order differences as additional features, which could provide us more information for user authentication. Therefore, a motion classifier based on motion features can be trained.

Figure 10 and Figure 11 illustrate the linear acceleration feature extracted by the accelerometer sensor along the Y-axis and the angular velocity feature extracted by the gyroscope sensor along the Z-axis, respectively, for the registered signature, the genuine signature, and the forged signature of one certain user. As can be seen in these two figures, the registered signature data are much closer to those of the genuine signature data. The forged signature data provided by a skilled attacker can be distinguished from the original user’s signature (both registered and genuine) using the decision model, which will be discussed in the following subsection.

### 3.3. Decision Model

In this subsection, we are going to discuss the decision model of the AirSign system.

#### 3.3.1. Architecture of the Decision Model

The flow chart of the decision model of the AirSign system is shown in Figure 12. The idea of the decision model is to organize the three types of classifiers—the hand geometry classifier, signature classifier, and motion classifier—as a combined cascade classifier.

#### 3.3.2. Hand Geometry Classifier

In the hand geometry section, the KNN-based classifier will compare the Euclidean distance between the hand geometry feature of the new “Air Signature” and the registered, genuine, and forged signature data. In our system, we chose K=33 for the KNN-based model. For each new “Air Signature”, it has 20 hand geometry samples and will pass the hand geometry classifier as long as the number of hand geometry samples classified into the positive data class is larger than a threshold. If this threshold is small, more forged signatures will pass the classifier, while a large threshold will cause more genuine signatures to be rejected. For AirSign, we use the number of 8 hand geometry samples as our threshold.

#### 3.3.3. Motion and Signature Classifiers

The dynamic time warping (DTW) method is a well-known technique to find an optimal alignment between two given sequences. This algorithm will return a DTW distance for each pair of sequences to determine the similarity. The method is used to compute the warping distance between the pair of sequences. Suppose that the input observation sequence is represented by w(i), where i=1,⋯,m, and the reference sequence by r(j), where j=1,⋯,n. Then, the distance D(i,j) in the DTW method is defined as the minimum distance starting from the beginning of the DTW table to the current position (i,j):(1)$$D\left(i,j\right)=d\left(i,j\right)+min\left\{\begin{array}{c} D(i-1,j)\\ D(i,j-1)\\ D(i-1,j-1) \end{array}\right.,$$
where d(i,j) is the distance matrix and can be defined as $d\left(i,j\right)={\left(w\left(i\right)-r\left(j\right)\right)}^{2}$.

Since the DTW method only compares two sequences and finds the best path with the least global distance, we use an extension of the original method—multidimensional dynamic time warping (MD-DTW) [11]. The MD-DTW method is used to calculate DTW by synchronizing multi-dimensional series. In order to generalize the DTW method for multidimensional sequence alignment, the matrix distance will be calculated by using the vector norm between a pair of points. Here, w(k,i) now is the input series and r(k,j) is the reference series, where *k* is the *k*th dimension of the point, i=1,⋯,m, and j=1,⋯,n. The matrix distance d(i,j) in MD-DTW is defined as
$$\begin{array}{c} d\left(i,j\right)=\sum_{k=1}^{K}{\left(w\left(k,i\right)-r\left(k,j\right)\right)}^{2}, \end{array}$$
where *K* is the number of dimensions of one point and the distance D(i,j) calculation will still follow Equation (Equation 1).

For our AirSign system, we implement the MD-DTW method on the acoustic features and motion features as dimensions and calculate the least global distance for each pair of registered data samples for each user.

As mentioned in Section 2, each user is required to register his/her signature for *N* times during the enrollment phase. These *N* registered signatures are selected as the training samples for each user. In either the signature section or motion section, N∗(N−1)/2 distances could be formed between each pair of registered signatures. Next, we calculate the average and standard deviation of the above N∗(N−1)/2 distances for both sections, denoted as the $mean_{s}$ and $std_{s}$ in the signature section and $mean_{m}$ and $std_{m}$ in the motion section, respectively. Therefore, the personalized thresholds
$$\begin{array}{cc} threshold_{s} & =mean_{s}+k_{s}*std_{s}\\ threshold_{m} & =mean_{m}+k_{m}*std_{m} \end{array}$$
could be trained for the signature classifier and the motion classifier for each specific user, respectively, where $k_{s}$ and $k_{m}$ are parameters inside the signature classifier and the motion classifier, respectively. With a high value of $k_{s}$ and $k_{m}$, more forged signatures will be accepted. A low value of $k_{s}$ and $k_{m}$ will cause more genuine signatures to be rejected.

When testing a new “Air Signature”, we first compare it with other *N* registered signatures to obtain *N* distances for both signature and motion. Then, we compare the average of these *N* distances with the personalized threshold in each section that we trained before to determine whether this new “Air Signature” could pass the signature and motion classifiers or not. We adjust the parameters $k_{s}$ and $k_{m}$ to make sure genuine signatures have a high possibility of passing the signature classifier, and let the motion classifier make the final decision for all passed signatures.

## 4. Data Collection

Since there is no public smartphone-based dataset available for our system, to better understand how AirSign performs in the real world, we conducted experiments to collect users’ data. For this purpose, we created an Android application to collect users’ hand geometry, signature, and motion data. The interface of the designed app, which is able to control the flow of our experiments, is shown in Figure 13. In the following subsections, we will describe more details about how we conducted these experiments.

### 4.1. Data Collection System

All experiments were conducted on a Huawei P20 smartphone provided by us, which has both acoustic sensors (earpiece speaker and top/bottom microphones) and motion sensors (accelerometer and gyroscope sensors). Specifically, for each experiment, designed sound signals would be transmitted by the earpiece speaker, travel through the air, reach the user’s hand, and be reflected back to the top and bottom microphones. Meanwhile, motion sensors would record the slight movement of the hand holding the smartphone.

For today’s smartphone, the common sampling frequencies that are used by acoustic sensors are 44.1 and 48 kHz. To obtain more precise in-air signature trace measurements, a sampling rate of 48 kHz was used for both the earpiece speaker and top/bottom microphones on the Huawei P20. The starting and finishing times of acoustic recordings for each signature were stored as Android system timestamps [16]. The Huawei P20 also includes both an accelerometer sensor with a sampling rate of 100 Hz and a gyroscope sensor with a sampling rate of 500 Hz. We saved the hand geometry and signature data from the acoustic sensors as an individual “.pcm” file with a size of 1–2 MB, while we stored each user’s motion measurements from the motion sensors as a “.txt” file with a size of 0.05–1 MB.

### 4.2. Data Collection Experiments

Our data collection process contained three sessions; 30 participants (20 male and 10 female) attended the first session, while 10 participants (seven male and three female) randomly selected from the above 30 participants joined both the second and third sessions. All participants held the Huawei P20 smartphone provided by us to sign in-air signatures through our designed Android application in our research lab. We also provided a survey to all participants asking about their experience of using AirSign. It took us about two weeks to collect all the data and we provided a 10-dollar gift card for each hour that they participated in the study as an incentive. The following experiments were approved by the institutional review board (IRB).

In the first session, 30 participants (undergraduate/graduate students at our institution) were asked to come every other day for three separate days to provide their registered signatures, genuine signatures, and forge other participants’ signatures. On Day 1, each participant was asked to select a name from a provided list of names (2–5 characters) for privacy concerns and practice signing this name in the air through our designed Android application for five minutes, with one hand holding the smartphone in portrait (vertically) and the other hand signing in the air. All participants were asked to sign over the smartphone screen in a three-dimensional space of 25 cm × 25 cm and 5–15 cm hovering over the earpiece speaker. Participants needed to practice first to meet our space requirements before the real experiment. Once a participant became proficient in signing the selected name in the air, he/she was then asked to register this signature 10 times. For each “air signature” in the registered phase, the signing hand/finger was held in front of the smartphone screen before signing for 2 s to collect hand geometry data (100 samples). In the meantime, we recorded how each participant signed his/her in-air signatures from different perspectives through a smartphone’s camera to imitate the shoulder surfing attacks. After registration, 10 genuine signatures were then collected from each participant. Only 0.5 s (20 samples) was needed for holding in front of the screen before signing the signature in the air this time. On each Day 3 and Day 5, 10 more genuine signatures were collected from each participant. In addition, we asked each participant to forge 10 signatures of two other randomly selected participants (five signatures each) by learning from the recorded videos. Participants could watch the recorded videos and practice as many times as they want before the attack. Hand geometry, as well as the signing process, would be carefully imitated so that these forgeries could have a high quality inside our database. Each participant had a total of 10 forged signatures after the random selection. Details about how to separate the above hand geometry data into training and testing datasets in our KNN-model have been described in Section 3.2.1.

In the second session, 10 participants were asked to sign signatures in different poses—standing, sitting, and lying on a chair or sofa. Ten registered, 10 genuine, and 10 forged signatures were collected under the same “quiet” environment from each participant within one day. At the training stage, all registered signatures were collected with a sitting pose, while genuine and forged signatures were collected in different poses. To evaluate the robustness of our method, at the test stage, 10 genuine and 10 forged signatures were collected in three different poses (sitting, standing, and lying) from each participant within one day. The procedure was the same as the one in the first session.

In the third session, 10 participants were asked to sign signatures in different environments (the “quiet” environment was a silent indoor environment with only a 40 dB sound pressure level measurement, the “talk” environment (50 dB) was an indoor environment with people talking at the same time, and the “music” environment (65 dB) was also an indoor environment with pop music being played at the same time). Ten registered, 10 genuine, and 10 forged signatures were collected with the same sitting pose from each participant within one day. At the training stage, all registered signatures were collected in a “quiet’’ environment, while genuine and forged signatures were collected in different environments. To evaluate the robustness of our method, at the test stage, 10 genuine and 10 forged signatures were collected in three different environments (quiet, talk, and music) from each participant within one day. The procedure was the same as the one in the first session as well.

In summary, our dataset is comprised of a total of 300 registered signatures, 900 genuine signatures, and 300 forged signatures from 30 participants in the first session and a total of 300 registered signatures, 300 genuine signatures, and 300 forged signatures from 10 participants in both second and third sessions. In total, 3300 signatures were collected from the three sessions.

## 5. Evaluation

This section discusses the performance results of AirSign from different perspectives.

### 5.1. How Well Does AirSign Perform Overall?

We use the first experiment session from Section 4.2 to demonstrate the overall performance using three possible systems—an acoustic classifier only, a motion classifier only, and both. The results are summarized in Figure 14.

As can be seen from this figure, the combination of acoustic and motion classifiers achieves the best F-score (97.1%), which is better than the F-score (92.4%) achieved by using an acoustic classifier only and the F-score (92.0%) achieved by using a motion classifier only. It can be observed that combining both the acoustic and motion classifiers will provide a more secure way to authenticate users because more distinguished features are extracted, which is hard to imitate for the attackers. Moreover, even without a motion classifier, our system still achieves an F-score = 92.4% using an acoustic classifier only, which also provides users another way to authenticate access to his/her smartphone without touching or holding the device. In addition, the motion classifier’s result also proves our assumption made in Section 1 that the human body will move as an entire entity when signing in the air in front of the smartphone. We also tested our system on a Google Pixel XL with 10 participants. After adjusting the sampling rates of the accelerometer sensor and gyroscope sensor, we obtained the final performance with a 96.3% F-score, which shows that AirSign has the applicability to work in the real world.

On the other hand, the average response delay of AirSign is 14.8 ms. The response delay is the time needed for AirSign to authenticate one signature after a user finishes signing his/her signature in the air. Since the acoustic signal processing is conducted simultaneously while a user is signing, the only part that causes response delay is calculating similarities using the MD-DTW method.

### 5.2. Will Users Be Able to Recall Their Signatures after a Few Days?

In this subsection, we again apply the first experiment session from Section 4.2 to test whether users could recall their signatures through AirSign after a few days’ rest. As can be seen in Figure 15, AirSign achieves 98.3%, 97.2%, and 95.9% F-scores using both the acoustic and motion classifiers on Day 1, Day 3, and Day 5, respectively. The performance is slightly degraded as the number of days increases because a user’s behavior is not always consistent on three different days, and he/she could wear different clothes or sign his/her signatures in different locations. In certain application scenarios, we may improve the performance by updating the registered signature database once a new signature passes the system.

### 5.3. Will Different Poses Affect the Authentication Accuracy?

In this subsection, we use the second experiment session from Section 4.2 to show the authentication accuracy of AirSign in three different poses—standing, sitting, and lying on a chair or sofa. Figure 16 shows the authentication performances in the above poses. As shown in this figure, AirSign performs the best with a sitting pose (97.5% F-score), while a lying pose has the lowest performance result (95.9% F-score) compared with the other two poses. With different poses, the data collected from the holding hand’s movement may be recorded differently, but they still perform well and do not influence the final authentication accuracy a lot by applying both the motion and acoustic classifiers. However, it is hard to do the authentication while a user is actively moving. More discussion is in Section 7.

### 5.4. Will Different Environments Affect the Authentication Accuracy?

In this subsection, we use the third experiment session from Section 4.2 to show the authentication accuracy of AirSign under three different environments—the “quiet” environment was a silent indoor environment, the “talk” environment was an indoor environment with people talking at the same time, and the “music” environment was also an indoor environment with pop music being played with the normal volume. The sound pressure levels measured in these three environments were 40, 50, and 65 dB respectively. Figure 17 shows the authentication performances in the above environments. We observe that the final results under these three different environments using both acoustic and motion classifiers do not change much by achieving 95.3%, 97.5%, and 94.9% F-scores in the quiet, talking, music environments, respectively. This is because AirSign uses higher frequency bands from 20 to 23 kHz, which could be separated from the audible sound noises by applying a Butterworth bandpass filter [15].

### 5.5. Why Was a KNN-Based Model Chosen for Classifying Hand Geometry?

As we mentioned in the previous section, in order to obtain useful hand geometry information from different users, a classifier must have a high recall such that it will not filter out any genuine data. Figure 18 shows four different machine learning models’ results. We compared KNN-based model with three other models—Random Forest, Naive Bayes, and SVM (support vector machine). From the data collection section, our final dataset for each participant included 200 positive training data, 600 positive testing data, and 200 negative testing data for the hand geometry classifier. Since we could not obtain a very large number of training examples, a neural network could not be applied in this case. As can be seen in this figure, the KNN-based model has the highest recall = 98.5%, while recall was only 89.9% for SVM, 76.2% for Naive Bayes, and 71.9% for Random Forest. According to the different properties of each machine learning model [17], KNN is insensitive to outliers, SVM is good at handling missing data, Random Forest is good at dealing with irrelevant features, and Naive Bayes is good for handling multiple classes. Combining all the above properties and the size of training examples, the KNN-based model fits our dataset and could provide us with the highest recall among these four machine learning models. Therefore, we selected a KNN-based model for classifying hand geometry. We adopted the Python machine learning package “scikit-learn” (http://scikit-learn.org/stable/index.html) to implement three machine learning methods: Random Forest (RF), K nearest neighbors (KNN), and support vector machine (SVM). For all these methods, we chose their hyper-parameters by conducting a five-fold cross validation.

### 5.6. How Do Hand Geometry, Signature, and Motion Classifiers Work for the Overall System?

In this subsection, we are going to show how these three classifiers work in our system. Table 1 and Table 2 provide the status information of genuine and forged data after the input signature data pass each classifier. In the beginning, our system does not know whether the input signature data are genuine or forged. Therefore, the “Unsure” rates for both the genuine and forged data are 100%. Finally, our system will output accurate rates for both kinds of data. The details of the two tables are discussed below.

In Table 1, the FRR (false recognition rate) represents the percentage of genuine users who are recognized as forged users, while the TAR (true acceptance rate) represents the percentage of genuine users who are recognized as genuine users.

In Table 2, the FAR (false acceptance rate) represents the percentage of the forged users who are recognized as genuine users, while the TRR (true recognition rate) represents the percentage of forged users who are recognized as forged users.

In the beginning, not all the input data are decided. After applying a hand geometry classifier, FRR = 1.8% in genuine data and TRR = 53.6% forged data are filtered out. Then, the FRR becomes 2.7% in the genuine dataset after passing the signature classifier, while 12.3% more forged data are filtered out. The motion classifier makes the final decision, and the final errors for the overall system are FRR = 3.4% and FAR = 1.6%. By implementing the above three classifiers in our dataset, the authentication performance is enhanced at each step and provides a more secure way to authenticate users.

### 5.7. How Many Registered Signatures Are Needed?

To test the relationship between the overall system performance and the size of registered signatures, we analyze AirSign on the data collected from the first experiment session with different numbers of registered signatures. Figure 19 shows the F-score of our proposed method under different numbers of registered signatures using both acoustic and motion classifiers. As can be seen in this figure, the F-score is relatively low given the small number of registered signatures (87.0% when the number of registered signatures is three) compared to other numbers of registered signatures. The F-score tends to be stable if more registered signatures are given (97.1% when the number of register signatures is seven). More registered signatures result in more comparisons and, hence, larger time consumption to complete the authentication process. Considering both the authentication performance and running time for AirSign, seven registered signatures are finally used.

### 5.8. How Do Participants Respond to AirSign?

All 30 participants answered the survey provided by us about the experience of using AirSign in terms of convenience, flexibility, and attractiveness. In terms of convenience, 27 participants thought the authentication process was easy to understand and that it was easy to use by signing signatures through AirSign. A total of 25 participants agreed with the flexibility because AirSign provided extra space to sign their signatures in the air compared with signing their signatures on the screen. For attractiveness, 29 participants thought AirSign was fascinating compared to other authentication methods, such as passwords, fingerprints, FaceID, etc., which means that AirSign is attractive to people.

### 5.9. How Does AirSign Compare to the Other Smartphone Authentication Systems?

Table 3 gives an overview of the existing methods for smartphone authentication methods from different perspectives. Among the commercial solutions, Samsung’s face recognition [6] is vulnerable to simple 2D image attacks and needs to be combined with other authentication methods for security [18]. Personal identification numbers (PINs) and graphical passwords [3] are the most popular smartphone authentication technologies; however, they are prone to shoulder-surfing attacks [4]. Apple’s TouchID and Samsung’s fingerprint are widely used and have good accuracy, but are vulnerable to finger mask attacks [19,20]. Apple’s FaceID [8] is the most secure method against 2D and 3D attacks owing to the TrueDepth camera system. However, Apple’s FaceID relies on extra-specialized hardware, which takes up a large space on the top of the screen. Among noncommercial methods, AirAuth [21] authenticates users according to in-air gestures, but it may require additional hardware, such as a short-range depth camera, which again takes a large space on the top of the screen. EchoPrint [22] leverages acoustics and vision for secure and convenient user authentication without requiring any special hardware, but the accuracy is low compared to our AirSign system. Z. Sitová et al. [23] introduced hand movement, orientation, and grasp (HMOG) to continuously authenticate smartphone users, but the accuracy is low compared to our AirSign system and needs additional hardware. SilentSign [24] and ASSV [25] both leverage acoustic signals to measure the distance variation of the tip of a pen while signing; however, they both rely on an additional 2D space to write the signature down. Compared to the above authentication methods, AirSign achieves a 97.1% F-score by using both built-in acoustic and motion sensors, which are readily available on most of today’s smartphones for user authentication. In addition, built-in motion sensors and microphones do not occupy the screen, and an earpiece speaker installed on the screen takes up a small space and may be moved out of the screen in the future. Moreover, AirSign can allow users to sign their signatures in a 3D space at any time and any place.

2

## 6. Related Work

We categorize existing works on (1) smartphone authentication, (2) signature authentication, and (3) acoustic sensing on smartphones.

### 6.1. Smartphone Authentication

Password authentication methods, such as personal identification numbers (PINs), or graphical passwords are the most natural and traditional smartphone user authentication technologies. However, PINs or graphical passwords can be easily acquired by other people [4]. Fingerprint authentication is pervasive in today’s smartphone authentication, but its accuracy is affected by the state of users’ fingers [7], and forging people’s fingerprints is possible [29]. More advanced fingerprint authentication technologies use ultrasonic signals [6] to capture the unique 3D characteristics of a user’s fingerprint, but this security method requires significant hardware changes to the smartphone. Apple’s FaceID [8] uses special sensors, such as a dot projector, a flood illuminator, and an infrared depth sensor, which require a large extra hardware cost.

In recent years, many solutions have been proposed for smartphone authentication research, which emphasize different behavioral biometric approaches [30] and use active sampling techniques. BreathPrint [31] senses the user’s breath sound using the in-built speaker and microphones to authenticate the user, which may have a significant impact when the user exercises intensively. Zheng et al. [32] utilized the accelerometer, gyroscope, and touchscreen sensors for non-intrusive authentication of a smartphone user by analyzing how a user touches the phone. However, due to user behavior changes, it may need the user to disable the verification function remotely and start the re-training to recognize these changes, which may cause a trade-off between security and convenience. Sitová et al. [23] introduced hand movement, orientation, and grasp (HMOG) to continuously authenticate smartphone users. The features of HMOG unobtrusively capture subtle micro-movements and orientation dynamics resulting from how a user grasps, holds, and taps on the smartphone. Karakaya et al. [33] used hand movement, orientation, and grasp (HMOG) sensor data to authenticate smartphone users as well. Yang et al. [34] performed a large-scale user study to collect a wide spectrum of signals on smartphones for signature authentication by involving multiple modalities on existing datasets, such as movement, orientation, touch, and gestures. More mobile authentications leveraging continuous multi-modal data [35,36] were proposed in mobile cloud environments. Differently from all the existing works, AirSign innovatively extracted unique in-air signature traces and hand geometries for smartphone authentication.

### 6.2. Signature Authentication

Generally, signature authentication can be summarized in two types: offline (static) and online (dynamic). Offline signature authentication uses solely 2D visual data acquired from scanning signed documents [37], while online signature authentication requires an electronic signing system, such as a digital tablet or iPad. The online signature authentication method provides dynamic features, like signature trajectory coordinates [38]. These dynamic features make online signatures more distinct and robust, as well as more difficult to be forged against offline signatures. Both the offline and online methods can potentially be used for real-time signature authentication, but in a real-world application, it is difficult to apply the offline methods, and most mobile devices’ signature authentication [39] methods associate the real-time capability with the online methods because of the greater diversity of signature features.

Among the overall processes of online signature authentication, which include data acquisition, pre-processing, feature extraction, and matching, feature extraction is the most essential part. Current approaches to online signature authentication can be divided into two categories: feature-based and function-based.

Feature-based: The calculation of a histogram is one of the feature-based approaches [40], which uses the histogram to compare the genuine and forged signatures on a smartphone. More specifically, it compares the extracted first-order and second-order differences of *x*, *y*, pressure, and angle, which are computed with respect to the horizontal axis as the features. Feature-based approaches also include the analysis of principal components [41]. However, Ref. [42] shows that homomorphic encryption can be easily applied to function-based methods, such as DTW, and thus, the feature-based approach no longer has a prominent security advantage over the function-based approach.

Function-based: Hidden Markov models (HMMs) are one type of function-based approach to solving online signature authentication problems. HMMs [43] are statistical Markov models that require a considerable number of reference signatures per user for training. An HMM-based online signature verification method is proposed in [44], which uses a set of time sequences and hidden Markov models on an electronic tablet. On the other hand, the dynamic time warping (DTW) method matches signatures directly with reference samples of the claimed user, which is another approach for solving online signature authentication problems. More precisely, DTW computes a dissimilarity score between two time sequences and outputs the warping path, which minimizes the score. A video-based system for in-air signature verification [45] uses DTW, FFT, and the analysis of the signature length to propose a video-based system, which enables smartphone users to complete the in-air signature verification process without touching the screen.

More advanced online handwritten signature authentication methods, such as SilentSign [24] and ASSV [25], adopt a device-free approach. SilentSign [24] leverages acoustic signals to measure the distance variation of the tip of a pen while signing. ASSV [25] uses a novel chord-based method to estimate phase-related changes caused by tiny actions and a trained deep convolutional neural network (CNN) model to verify signatures. Compared to the aforementioned works, AirSign novelly leverages both acoustic and motion sensors on the smartphone to detect hand geometry and a finger’s signing trace in a 3D space, which improves both the robustness and the security level of the authentication.

### 6.3. Acoustic Sensing on Smartphones

Acoustic sensing has been used for ranging, localization, gesture recognition, and tracking due to its slow propagation speed, which can improve the accuracy of measurement results. We classify existing work into a device-based approach and a device-free approach.

Device-based: Most existing acoustic-based recognition and tracking research is device-based. One part of this research uses acoustic signals to estimate distance. BeepBeep [46] designs and implements a high-accuracy acoustic-based ranging system that allows the sender and receiver with unknown clock offsets to measure the one-way propagation delay. Sword-Fight [47] develops a fast, accurate, and robust localization system that enables two potentially fast-moving phones to keep accurate distance estimates with each other. Another part of this research uses acoustic signals for localization. Cricket [48] uses both Radio Frequency (RF) and audio to achieve a median error of 12 cm with six beacon nodes. Shake and Walk [49] exploits the Doppler shift in an audio signal for fine-grained indoor localization. AAmouse [50] uses Doppler shifts to track a phone’s position using anchor devices in a room. CAT [51] leverages external speakers and uses frequency-modulated continuous waves (FMCWs) for phone movement tracking at millimeter-level accuracy. The third part of acoustic-based approaches is aimed at gesture recognition. All of these approaches can only deal with predefined gestures and are not designed for the same gesture from different people.

Device-free: ApneaApp [52] tracks the periodic breathing movements using FMCW reflections of the inaudible transmissions from smartphones. LLAP [12] develops a device-free gesture-tracking scheme using the acoustic phase to get fine-grained movement direction and movement distance measurements. FingerIO [53] explores the feasibility of using commercial mobile devices to track fingers and hands within a short distance. BatMapper [15] uses acoustics for fast, fine-grained, and low-cost indoor floor plan construction.

## 7. Discussion and Future Work

### 7.1. Users’ Active Motion

Currently, it is hard for AirSign to authenticate a user when he/she is actively moving (i.e., walking, running, climbing, driving, etc.). These motions captured by the accelerometer and gyroscope sensors will inevitably cause dissimilarity between registered signatures and input signatures. Currently, we only tested our system while the users were static (standing, sitting, and lying on a chair or sofa). The holding hand’s motion may be extracted if the moving pattern of the user is learned by using machine learning tools. This will add much more complexity to our system, and we leave it for future work.

### 7.2. Multiple Users

If multiple users are trying to authenticate using AirSign in the same place at the same time, the transmitted and reflected signals with the same frequency interval from different smartphones may interfere with each other. Consequently, authentication performance may be affected. In the future, we plan to explore the feasibility of dynamically choosing a lower number of frequency bands or different frequency intervals to address the above frequency collision problem and allow multiple users to authenticate through AirSign simultaneously.

### 7.3. Privacy Concerns and Memory Concerns

All of our signature data are anonymous, and all the users have already signed the data publishing contract. For international business purposes, multiple data encryption technologies, such as RSA (Rivest, Shamir, and Adleman), AES (Advanced Encryption Standard), and DES (Data Encryption Standard) could be easily adopted to store the users’ data.

Note that one user’s data, which consist of many signatures, only take around 1 MB. It is typical for a modern smartphone to have a memory of at least 32 or 64 GB. Therefore, at the current stage, memory cost should not be the bottleneck of our method. As time goes by, a user’s genuine signature may consume more and more memory. We will set a reasonable memory cost threshold, which leads to a good authentication accuracy based on the memory size of the smartphone. We only need to use the new data to update the old data after reaching the memory threshold.

### 7.4. Large-Scale Experiment

We only had 30 users in the current experiments and 10 users for the subsequent experiments (on pose and environment), while large-scale experiments with more than 100 or even more users for each experiment are needed for a better and more mature solution. In the future, we will seek ways to do experiments at such a large scale.

## 8. Conclusions

In this paper, we proposed AirSign, which enables users to sign in the air for signature authentication. The system leverages acoustic and motion sensors of smartphones without using extra hardware and authenticates users as being genuine if their signatures pass all three classifiers—hand geometry, signature, and motion classifiers. The experiments showed that AirSign is able to distinguish genuine and forged data with a 97.1% F-score and is convenient and flexible to use.

## Figures and Tables

**Figure 1 sensors-21-00104-f001:**
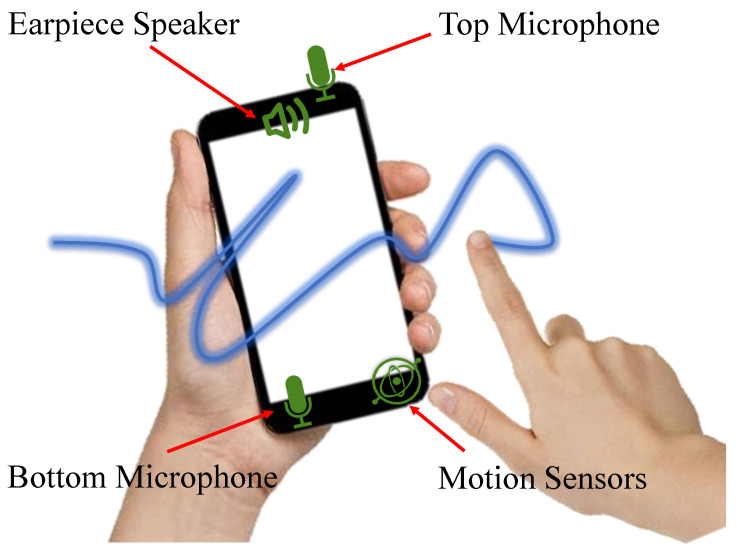
AirSign leverages both motion and acoustic sensors of smartphones to extract hand geometry, signature, and motion features while the user is signing his/her signature in the air.

**Figure 2 sensors-21-00104-f002:**
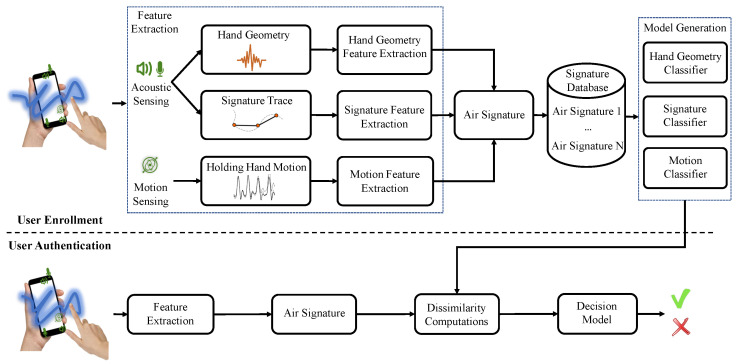
AirSign system architecture—user enrollment and user authentication.

**Figure 3 sensors-21-00104-f003:**
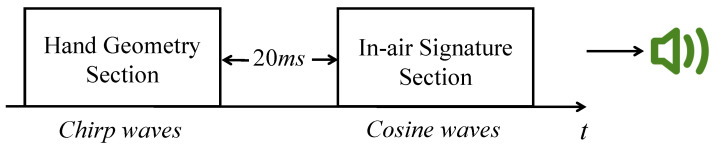
Sound signal design overview.

**Figure 4 sensors-21-00104-f004:**
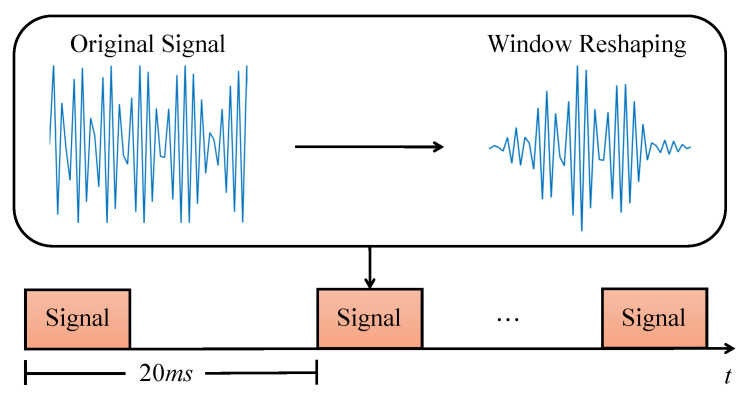
Hand geometry transmitter design.

**Figure 5 sensors-21-00104-f005:**
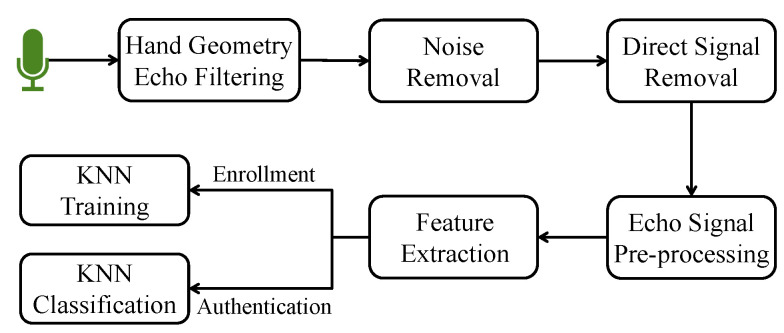
Hand geometry receiver design.

**Figure 6 sensors-21-00104-f006:**
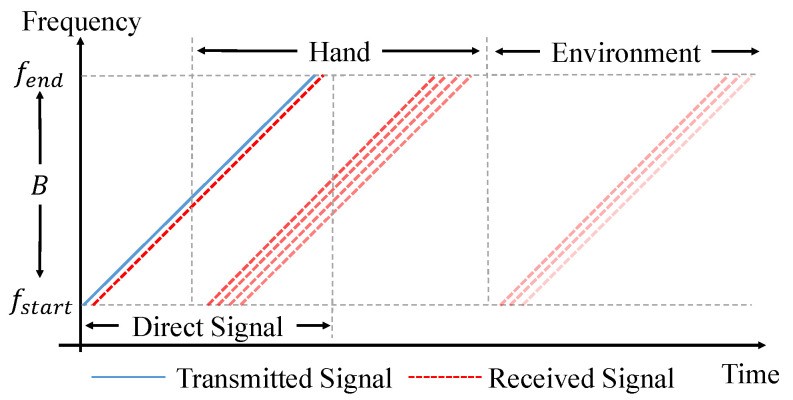
Relationship between time and frequency of the transmitted signal and received signals.

**Figure 7 sensors-21-00104-f007:**
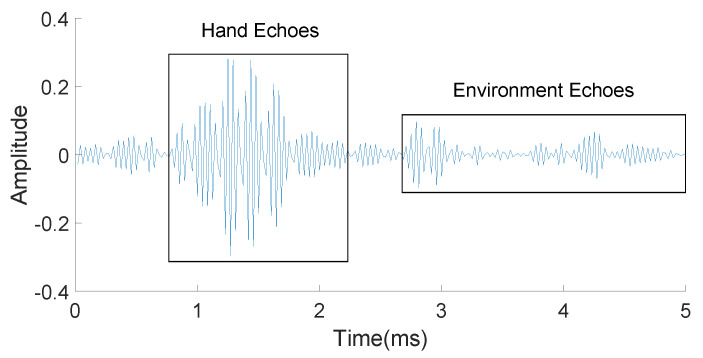
Separation of hand echoes and environment echoes.

**Figure 8 sensors-21-00104-f008:**
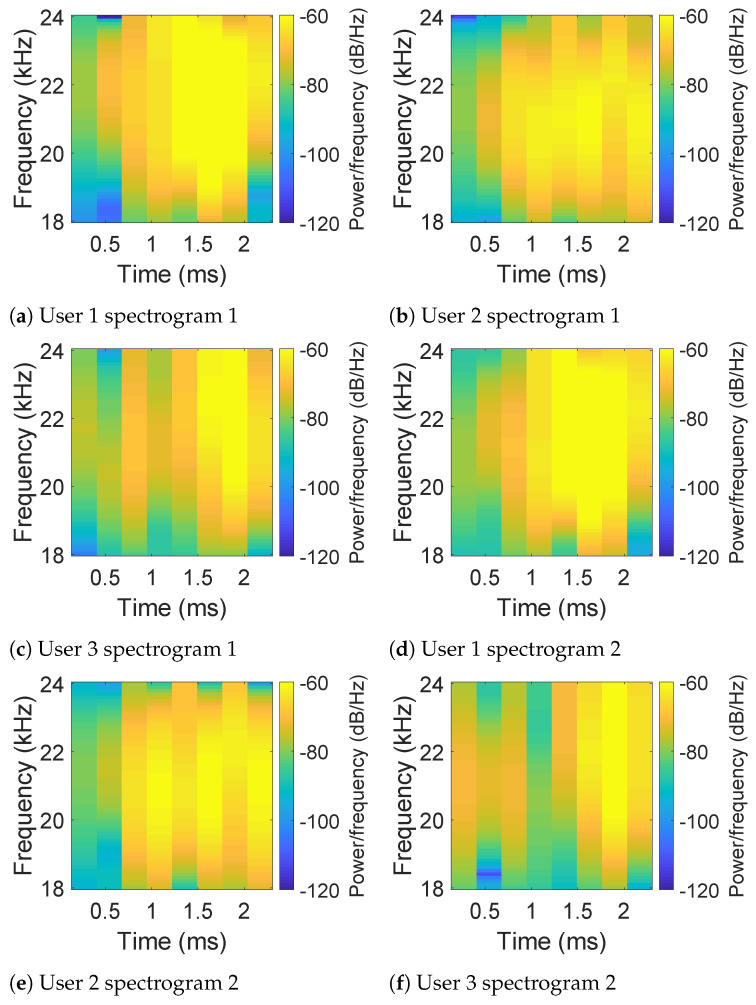
Spectrograms of echoes obtained by the top microphone in the hand geometry section for three different users.

**Figure 9 sensors-21-00104-f009:**
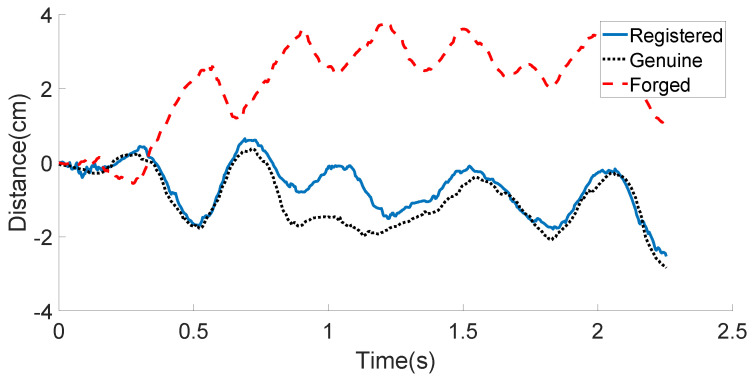
User’s bottom microphone displacement example: registered, genuine, and forged in-air signatures.

**Figure 10 sensors-21-00104-f010:**
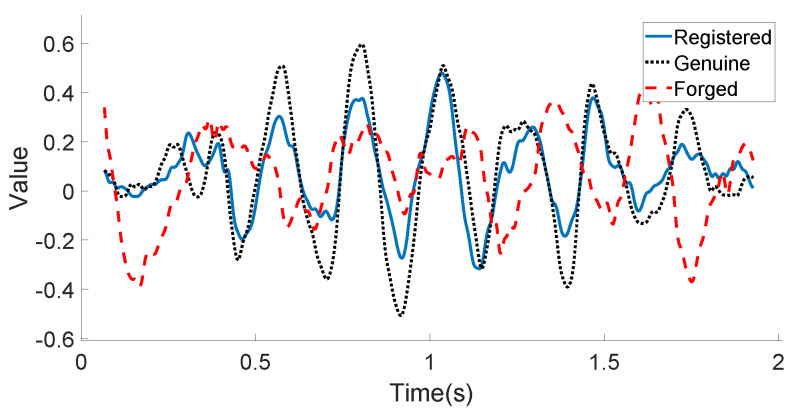
User’s motion sensor data example (Acceleration-Y): registered, genuine, and forged in-air signatures.

**Figure 11 sensors-21-00104-f011:**
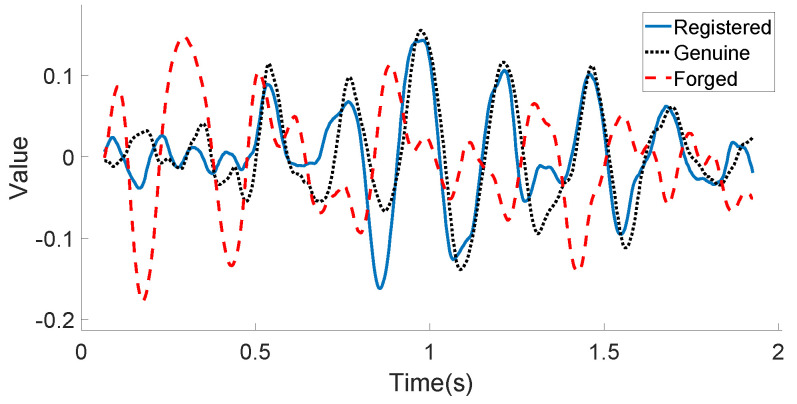
User’s motion sensor data example (Gyroscope-Z): registered, genuine, and forged in-air signatures.

**Figure 12 sensors-21-00104-f012:**
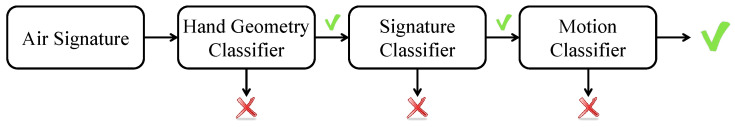
Flow chart of the decision model.

**Figure 13 sensors-21-00104-f013:**
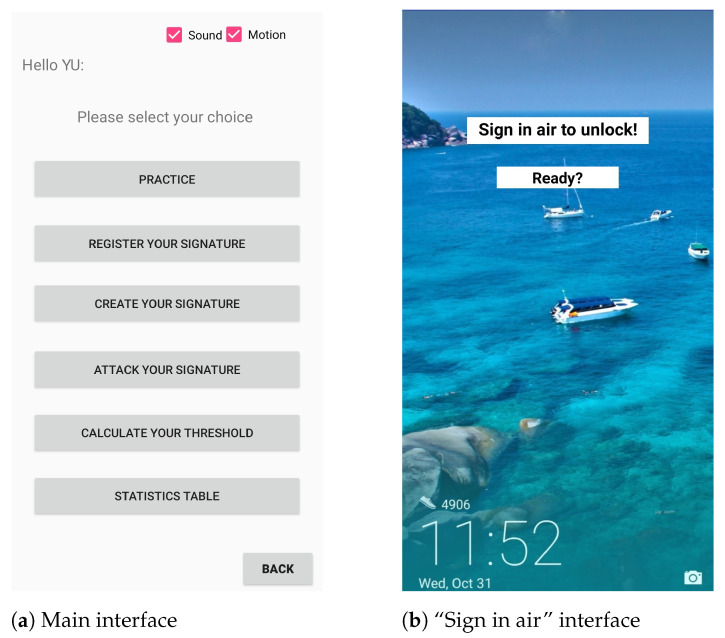
Data collection application user interface (UI).

**Figure 14 sensors-21-00104-f014:**
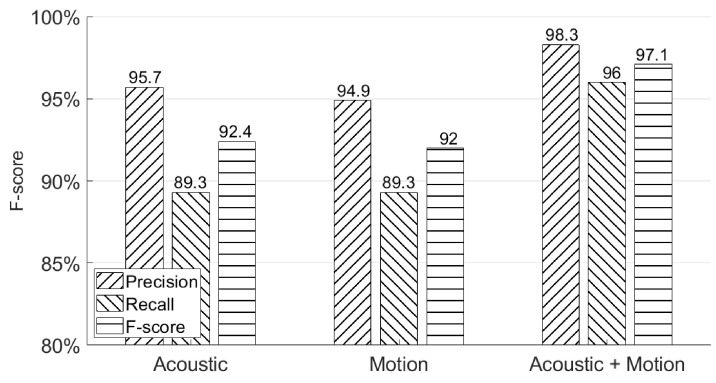
Overall system authentication performances.

**Figure 15 sensors-21-00104-f015:**
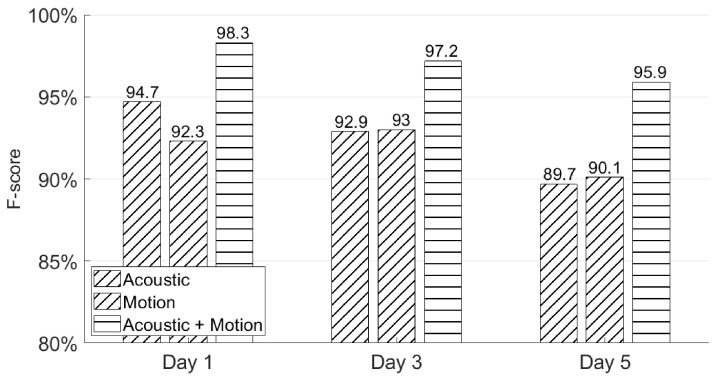
Different days’ authentication performances.

**Figure 16 sensors-21-00104-f016:**
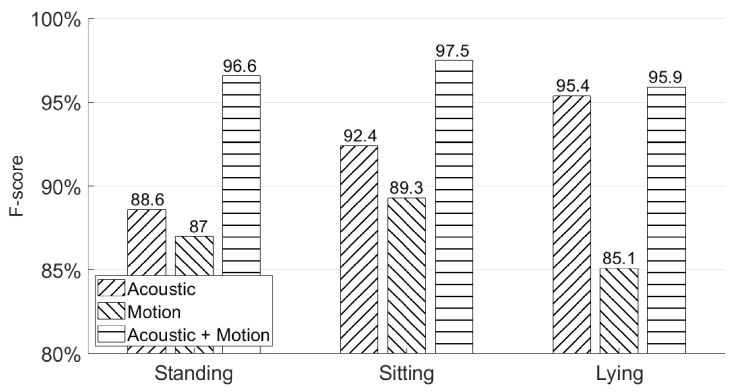
Different poses’ authentication performances—standing, sitting, and lying.

**Figure 17 sensors-21-00104-f017:**
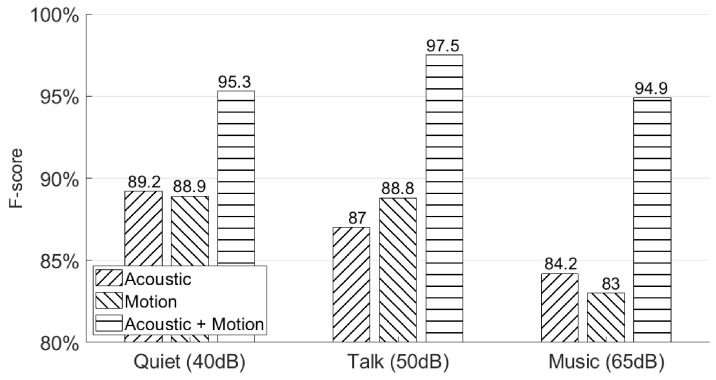
Different environments’ authentication performances—quiet, talking, and music.

**Figure 18 sensors-21-00104-f018:**
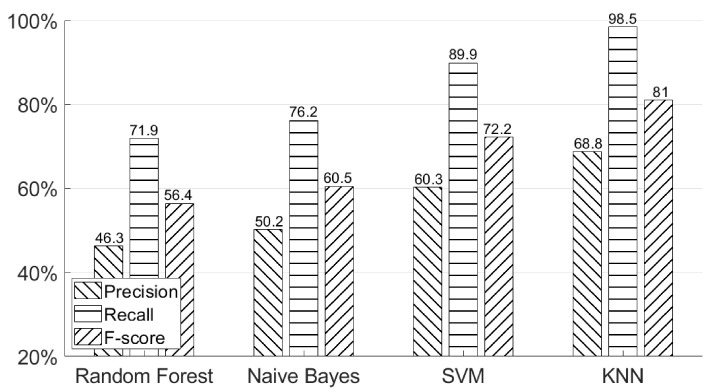
Authentication performances using different machine learning models—Random Forest, Naive Bayes, SVM and KNN.

**Figure 19 sensors-21-00104-f019:**
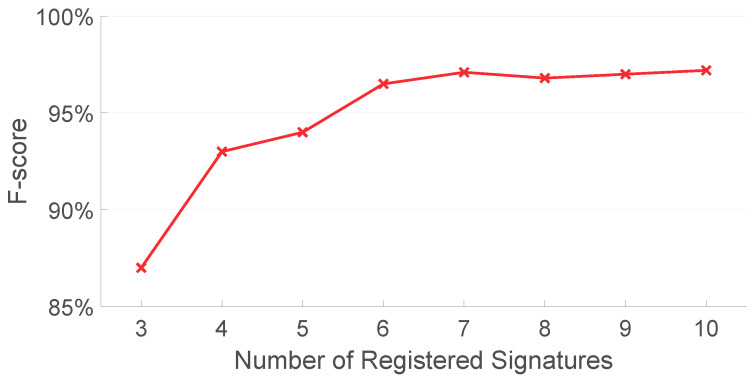
Authentication performances with different numbers of registered signatures.

**Table 1 sensors-21-00104-t001:** Statistic table for genuine data after each step of the classifiers.

	Unsure (%)	FRR (%)	TAR (%)
Input	100	0	0
Hand Geometry Classifier	98.2	1.8	0
Signature Classifier	97.3	2.7	0
Motion Classifier	0	3.4	96.6

**Table 2 sensors-21-00104-t002:** Statistic table for forged data after each step of the classifiers.

	Unsure (%)	FAR (%)	TRR (%)
Input	100	0	0
Hand Geometry Classifier	46.4	0	53.6
Signature Classifier	34.1	0	65.9
Motion Classifier	0	1.6	98.4

**Table 3 sensors-21-00104-t003:** Summary of existing smartphone authentication methods.

Technique	Hardware	Screen Space	Limitation	Accuracy
		Occupied		
*FaceID [8]*	TrueDepth camera	Large	3D head mask attack[26]	>99.9%
*Samsung FR [27]*	RGB camera	Medium	Images attack [18]	-
*TouchID [5]*	Fingerprint sensor	Large	Finger masks [20]	>99.9%
*Samsung FP [6]*	Ultrasonic fingerprint sensor	None	Finger masks [19]	>99.9%
*PIN [3]*	Smartphone screen	None	Shoulder-surfing attack [4]	-
*AirAuth [21]*	Depth camera	Large	Additional hardware	EER 3.4%
*Z. Sitová et al. [23]*	Motion sensors and	Large	Low accuracy	EER 7.16% (walking)
	touch screen			EER 10.05% (Sitting)
*EchoPrint [22]*	Acoustic sensors and camera	Medium	Low accuracy in low illumination	93.5%
*SilentSign [28]*	Acoustic sensors	Small	Handwritten signature by pen	EER 1.25%
*ASSV [25]*	Acoustic sensors	Small	Handwritten signature by pen	EER 5.5%

## Data Availability

The data presented in this study are available on request from the corresponding author. The data are not publicly available due to privacy and ethical.

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
