# Peer review of "AirSign: Smartphone Authentication by Signing in the Air"

_sensors, 2020, doi:10.3390/s21010104_

Round 1

Reviewer 1 Report

This paper proposed an acoustic-based handwritten signature tracking and authentication method, which sensed both the static hand geometry and dynamic signature trajectory. The authors employed two kinds of inaudible acoustic sensing techniques, i.e., the chirp-based spectrogram, and phase change-based tracking, to capture the hand geometry and signature trajectory respectively. Besides, the authors also used motion sensors to capture the subtle movements of another hand that held the smartphone as complementary factors to enhance the user authentication.

Overall, the idea of tracking the handwritten in-air signature for authentication is not new, but the authors consider various factors to build a new biometric, which looks interesting. And the paper is easy to follow. However, considering the lacked insights, straightforward techniques, this paper is far from publication. Detailed comments are as follows.

There have already existed various researches on exploring in-air signature as a factor for authentication using acoustic sensing techniques [1][2]. What is the difference between this work and existing works? The authors only emphasize the advantage of the in-air signature compared with traditional biometric-based authentications, which is in sufficient. Considering this point, the contribution of this paper is limited.

[1] Mengqi Chen, Jiawei Lin, Yongpan Zou, Rukhsana Ruby, Kaishun Wu: SilentSign: Device-free Handwritten Signature Verification through Acoustic Sensing. PerCom 2020: 1-10, 2019

[2] Feng Ding, Dong Wang, Qian Zhang, Run Zhao: ASSV: Handwritten Signature Verification Using Acoustic Signals. Proc. ACM Interact. Mob. Wearable Ubiquitous Technol. 3(3): 80:1-80:22, 2019

Many insights underlying the in-air signature-based authentication are lacked. What is the core of distinguishing different persons using in-air signature? The signature itself? Or the underlying behavioral characteristics? A more specific example is that the authors proposed to involve the subtle movements of the holding hand using motion sensors to be an authentication factor. However, why could the movements be an authentication factor? In my opinion, such a subtle movement could not exhibit the behavioral characteristics, even though the authors showed the authentication result based on motion sensors is good.

Both acoustic techniques are straightforward, limiting the technical contributions of this work. The acoustic sensing occupied most spaces of this paper, thus being considered as the main technique in this work. However, both techniques, i.e., chirp-based spectrogram (really similar to FMCW, even simpler), and phase-based tracking (directly from the literature [10]), are without technical elaboration. It is fine to directly employ existing techniques on an engineering work, but not good in a research paper. The authors should consider the specific problems encountered in this topic, and provide unique solutions (even build on existing techniques), so as to improve the contributions.

In Section 5.3, the authors provide the performance evaluation under different poses. I am curious that what the pose during training is. Is the pose the same with testing? If so, I do not agree the proposed system could achieve the robustness on different poses. The authors should provide more details of this experiment for a clearer understanding.

Reviewer 2 Report

This paper presents a novel multi-biometric verification system that asks users to provide their sign in the air close to the smartphone screen. To this end, the authors extract acoustic features from user's hand geometry and sign gesture, by leveraging built-in microphones and speakers. In addition, they compute unique characteristics of the user from the data collected by built-in motion sensors (e.g., accelerometer and gyroscope). These features are used to train three classifiers that provide a decision on whether the features come from the expected user. Experiments conducted on data from 30 participants showed that combining acoustic and motion features leads to an effective recognition. 

The paper presents a nice and promising biometric authentication system, which provides interesting findings. One point I particularly liked of this paper is the originality of the proposed authentication system. However, I have some key concerns regarding the methodology and design choices made by the authors, that should be addressed. In what follows, I detail my main concerns.

- Ethical and practical issues of the classifier training.
The decision model described in Section 3.3 requires both genuine and impostor data for being trained successfully. If the proposed authentication system is deployed on a smartphone, this means that a smartphone of a given user will need to store data of other users as well. First, there might be several privacy concerns, due to the national/international regulations. One strategy to deal with this problem is the use of one-class classifiers which are trained only on data from the genuine user to identify anomalies with respect to it, during the authentication phase. Second, due to memory constraints, storing a large dataset of impostors on a smartphone might be challenging. Hence, questions such as how many impostors are need to reasonably train a classifier and if the impostor population is adequately representative still remain unanswered.

- Generalizability of the findings.
I really appreciated the work made by the authors to collect the data and how the experimental section is presented. However, while findings are promising, the number of users (30) included in the first experiment left the reader (and I) wondering whether such findings generalize to a larger population. Moreover, the subsequent experiments (on pose and environment) further reduce the population, making this issue more evident. Another weak point of the selected population is that it is composed only of undergraduate/graduate students at the authors' institution, so the participants cover only one small subpopulation of future potential users of this biometric system. 
Given that the proposed system relies on key elements of the smartphone and that the authors used the same unique smartphone for all the experiments, it remains unclear whether the same findings hold, when other smartphones are used. Motion and acoustic sensors can greatly vary (eg., in terms of hardware) across devices, so more smartphone models should be included in the experiments to better assess the applicability of this method in the real world.
The authors made a good job in showing the results for both uni- and multi-biometric setups. However they should present the effectiveness of the uni-biometric acoustic submodule when data comes from the hand-geometry step and the signature step, separately. Currently, it is not clear whether the distinctiveness of the acoustic patterns comes from one step or the other, or from both. In addition, the authors should include the Equal Error Rate of the considered settings, to foster comparability with other authentication systems.

Reproducibility and replicability of the results.
There is no enough information about the experimental conditions, e.g., on the specific details of the talking and music environments. Further, the authors did not share the code, and this limits the possibility of verifying results and authors claims by reviewers in other ways. There is also no description of how the authors tuned the models. Overall, it appears challenging for a researcher to reproduce the paper methodology and the results, starting from the content in the paper, given also that the datasets is not publicly available. This means that other researchers may not be able to reproduce insights or use them in any way that allows them to advance in different directions. Reproducibility (and if possible replicability) are two key aspects of research, and they should be carefully considered in a journal paper which expects to have high impact.

- Comparison with respect to other strategies.
The comparison provided in Section 5.7 is weak and not adequately supported by appropriate references. Given that the method proposed by the authors is new, it is likely that there exist no extensive attacks against it, differently from other widely adopted methods such as PIN or AirAuth. I believe that the authors' claims in favor of their system are a bit over-estimated. Therefore, the mentioned section requires a large revision in order to correctly reflect the state of the current research and highlight pros and cons of the alternatives.

- Connection to the related work.
Smartphone authentication by means of biometrics has been studied for a long time. However, the authors only include a very narrowed set of references, which fails to cover an enough representative view of the state of the art. Therefore, the authors should expand their discussion in such related work, supporting it with appropriate (at least) recent references. Current literature, such as https://link.springer.com/chapter/10.1007/978-3-319-70742-6_31 - https://www.sciencedirect.com/science/article/pii/S1877050919313845 - https://ieeexplore.ieee.org/document/8436074 - https://ieeexplore.ieee.org/abstract/document/7349202 - https://link.springer.com/chapter/10.1007/978-3-030-58951-6_35 - https://dl.acm.org/doi/10.1145/2668332.2668366, should be adequtely discussed in order to place this paper in the context and better highlight the research gap. Similar points apply to the other related work sections. 

Round 2

Reviewer 1 Report

After a round of revision, the authors have addressed most of my concerns. But some more in-depth insights should still be given. In particular, the authors claimed that they consider the human body an entire entity, so that the acoustic and motion sensors capturing both hands’ behaviors could be used for authentication. However, there is no supporting evidence verifying this conclusion. Hence, I am not convinced by the claims. I suggest the authors to find some literatures from biology or medicine fields to support their conclusions.

Reviewer 2 Report

I have appreciated the changes made by the authors. I have found that they have addressed my comments and the paper is now ready to be accepted, since it has improved in ethical and technical issues, generalizability, reproducibility,  and connection with prior work. 

Author Response

We are grateful to know that all concerns have been addressed. We appreciate your comments and suggestions for our paper.